# New Functions of Intracellular LOXL2: Modulation of RNA-Binding Proteins

**DOI:** 10.3390/molecules28114433

**Published:** 2023-05-30

**Authors:** Pilar Eraso, María J. Mazón, Victoria Jiménez, Patricia Pizarro-García, Eva P. Cuevas, Jara Majuelos-Melguizo, Jesús Morillo-Bernal, Amparo Cano, Francisco Portillo

**Affiliations:** 1Departamento de Bioquímica UAM, Instituto de Investigaciones Biomédicas Alberto Sols, CSIC-UAM, 28029 Madrid, Spain; peraso@iib.uam.es (P.E.); mazonmaria@gmail.com (M.J.M.); victoria.jimenez@uam.es (V.J.); patripizgar@gmail.com (P.P.-G.); e.p.cuevas@csic.es (E.P.C.); jara_mm@hotmail.com (J.M.-M.); jmorillo@iib.uam.es (J.M.-B.); amparo.cano@inv.uam.es (A.C.); 2Instituto de Investigación Sanitaria del Hospital Universitario La Paz-IdiPAZ, 28029 Madrid, Spain; 3Centro de Investigación Biomédica en Red, Área de Cáncer (CIBERONC), Instituto de Salud Carlos III, 28029 Madrid, Spain

**Keywords:** LOXL2, intracellular LOXL2, nuclear interactome, RNA-binding proteins, EMT

## Abstract

Lysyl oxidase-like 2 (LOXL2) was initially described as an extracellular enzyme involved in extracellular matrix remodeling. Nevertheless, numerous recent reports have implicated intracellular LOXL2 in a wide variety of processes that impact on gene transcription, development, differentiation, proliferation, migration, cell adhesion, and angiogenesis, suggesting multiple different functions for this protein. In addition, increasing knowledge about LOXL2 points to a role in several types of human cancer. Moreover, LOXL2 is able to induce the epithelial-to-mesenchymal transition (EMT) process—the first step in the metastatic cascade. To uncover the underlying mechanisms of the great variety of functions of intracellular LOXL2, we carried out an analysis of LOXL2’s nuclear interactome. This study reveals the interaction of LOXL2 with numerous RNA-binding proteins (RBPs) involved in several aspects of RNA metabolism. Gene expression profile analysis of cells silenced for LOXL2, combined with in silico identification of RBPs’ targets, points to six RBPs as candidates to be substrates of LOXL2’s action, and that deserve a more mechanistic analysis in the future. The results presented here allow us to hypothesize novel LOXL2 functions that might help to comprehend its multifaceted role in the tumorigenic process.

## 1. Introduction

LOXL2 belongs to the lysyl oxidase (LOX) protein family, which is constituted by five members: LOX, and four LOX-like enzymes (LOXL1-4). All members are characterized by a conserved carboxyl (C)-terminal amine oxidase catalytic domain that includes a histidine-rich copper-binding motif and a lysyl-tyrosyl-quinone cofactor, both of which are essential for the catalytic activity [1]. By contrast, the amino (N)-terminal region diverges among all members. In the case of LOXL2-4, they present four scavenger receptor cysteine-rich (SRCR) domains. Based on N-domain diversification and sequence comparison, the LOX protein family has been classsified into two subfamilies: one constituted by LOX and LOXL1, and the other including LOXL2, LOXL3, and LOXL4 [2,3]. The LOXL2-4 SRCR domains are identical to those that are widely distributed among members of the scavenger receptor superfamily [4]. The functional role of LOXL2’s SRCR domains has not yet been well characterized. It is assumed that they could be involved in protein–protein interactions by analogy to the function of the SRCR domains present in other members of the scavenger receptor superfamily [5,6]. Nevertheless, it has been recently observed that specific LOXL2 and LOXL3 SRCR domains are able to deacetylate and deacetyliminate multiple acetyl-lysine residues of the STAT3 transcriptional factor by themselves [7]. Moreover, it has recently been described that the LOXL2 splice variant L2Δ13, which lacks the amino oxidase domain, is able to directly catalyze the deacetylation of aldolase A at Lys-13 [8]. These results may imply a new function of the SRCR domains that has not been described so far.

The canonical function of LOX enzymes is the maturation of the extracellular matrix (ECM). LOX enzymes catalyze the oxidative deamination of ε-amino groups of peptidyl-lysine and hydroxylysine residues to produce highly reactive aldehydes that undergo a spontaneous condensation, thereby establishing intra- or inter-crosslinkages in collagen and elastin [9,10]. Despite their well-established role in ECM maturation, lysyl oxidase proteins are also associated with diverse physiological and pathological processes, including fibrosis, cancer, and cardiovascular diseases (as reviewed in [9,10,11,12,13]).

Regarding LOXL2, increasing evidence has accumulated showing its role in several human cancers [14,15,16]. Interestingly, the role of LOXL2 in this pathological context has been associated with its intracellular location. Our previous studies in a large series of human tumor samples demonstrated that intracellular LOXL2 is associated with poor prognosis in laryngeal squamous-cell carcinomas and with distant metastasis in basal-like breast carcinomas [17,18]. Mechanistically, intracellular LOXL2 promotes tumorigenesis through transcriptional and posttranscriptional actions impinging on epithelial-to-mesenchymal transition (EMT)—a key process in the metastatic cascade. EMT is a genetic and reversible program that leads to the loss of epithelial status, apical–basal polarity, and cell–cell adhesions, as well as to the gain of mesenchymal traits, resulting in cells with a greater capacity for mobility, migration, and invasion [19]. Several transcription factors (TFs) have been described during the past decade as EMT inducers (EMT-TFs). A plethora of signaling pathways converge in the activation of one or more of the core EMT-TFs, including the zinc finger TFs of the SNAIL (SNAI1, SNAI2) and ZEB (ZEB1 and ZEB2) families, and the basic helix-loop-helix TFs TCF3 (also known as E47) and TWIST1 [20,21,22,23]. LOXL2 induces EMT through several mechanisms. It collaborates with SNAI1 to repress CDH1 (E-cadherin gene) expression by counteracting the SNAI1 GSK3β-dependent degradation, thereby increasing SNAI1’s stability [24]. LOXL2 can also oxidize the trimethylated lysine 4 in histone 3 (H3K4me3) after SNAI1 binding to heterochromatin, thereby repressing CDH1 transcription [25,26]. Repression of E-cadherin expression is also mediated by the interaction between LOXL2 and TCF3/E47 and their direct binding to the CDH1 promoter region [27]. LOXL2 can also mediate the transcriptional downregulation of components of tight junctions and cell polarity complexes, including the *claudin-1* and *Lgl2* genes [18]. Moreover, in skin cancer cells, LOXL2 collaborates with KLF4 in the NOTCH1 promoter, where it oxidizes H3K4me3, thereby impairing RNA polymerase II recruitment and inhibiting NOTCH1 transcription, repressing epidermal differentiation [28]. More recently, we reported that LOXL2 overexpression promotes its accumulation in the endoplasmic reticulum, where it interacts with the HSPA5 chaperone, leading to the activation of the IRE1-XBP1s and PERK signaling pathways of the unfolded protein response, which, in turn, induces the expression of several EMT-TFs (SNAI1, SNAI2, ZEB2, and TCF3) [29]. Some of these actions are independent of LOXL2’s amino oxidase catalytic activity [7,8,29,30,31].

As described above, many studies show that LOXL2 promotes the progression of many types of tumors. This implies that inhibitors decreasing LOXL2 expression or activity may be useful therapeutic agents for the treatment of many types of cancer. The development of LOX2 inhibitors started with simtuzumab—a humanized LOXL2-targeting antibody that failed to show clinical effectiveness in several phase 2 clinical trials in several types of cancer [32,33]. Recently, efforts have focused on the search for small-molecular-weight LOXL2 inhibitors. One example is PXS-S1A—a first-generation LOXL2 inhibitor derived from haloallylamine that has shown promising results in preclinical models. In MDA-MB-231 human model cells of breast cancer, PXS-S1A inhibited the growth of primary tumors and reduced primary tumor angiogenesis, although it was less efficient at blocking the overall metastatic burden in the lungs and liver [34]. In any case, we must be aware that lysyl oxidase family members may have overlapping functions, as recently shown for LOXL2 and LOXL3 [35]. Additionally, it should be considered that small-molecular-weight LOXL2 inhibitors are designed to block the catalytic activity of the enzyme, and we know that the classical oxidase activity is not required for many of the intracellular functions of LOXL2 in cancer, such as the induction of EMT.

Despite the accumulated evidence for LOXL2’s intracellular functions, the mechanisms of LOXL2’s actions promoting tumor progression are not yet fully understood. With the aim of further understanding the intracellular LOXL2 functions impinging on tumorigenesis, we explored the nuclear interactome of LOXL2. To this end, we performed immunoprecipitation experiments on a nuclear-enriched fraction of cells expressing a flag-tagged version of LOXL2, and we analyzed the proteins co-immunoprecipitating with LOXL2 by LC-MS/MS. The quality of the interactome was evaluated by co-immunoprecipitation analyses of endogenous LOXL2 and selected interacting proteins. LOXL2’s nuclear interactome is composed of numerous RNA-binding proteins (RBPs) involved in all aspects of mRNA metabolism. The deletion of individual LOXL2-SRCR domains, combined with co-immunoprecipitation experiments, suggested that the SRCR-1 domain is required for LOXL2’s interaction with nuclear proteins. To identify, among the possible RBPs targeted by LOXL2, those functionally relevant in tumorigenesis, a gene expression profile of cells silenced for LOXL2 was performed, followed by a selection of genes directly involved in EMT. This EMT signature was further scrutinized for genes containing binding sites for the RBPs identified in the LOXL2 interactome. After this study, six RBPs (ELAVL1, FMR1, IGF2BP1, TAF15, SRSF1, and U2AF2) emerged as solid candidates to be regulated by LOXL2, with potential implications in EMT regulation.

## 2. Results

### 2.1. Nuclear Interactome of LOXL2

To identify LOXL2’s potential nuclear partners, we utilized HEK293T cells transfected with an empty flag vector or carrying a flag-tagged version of LOXL2. Nuclear-enriched extracts of duplicated cell cultures were immunoprecipitated with the corresponding anti-tag antibodies and subjected to LC-MS/MS identification. Prior to analysis, the hits identified in immunoprecipitates from flag-empty vector experiments were subtracted from the group of proteins immunoprecipitated with flag-LOXL2 (Appendix A). Proteins identified with two or more peptides in the two experimental flag-LOXL2 immunoprecipitates were selected, resulting in a core set of 107 proteins (Figure 1A). While HSPA5 and IRS4 were identified as LOXL2 interactors in a previous study [36], most of the identified proteins in the present nuclear LOXL2 interactome represented novel interactions. To assess the robustness of the obtained interactome, we analyzed the interaction of endogenous LOXL2 with selected interactome proteins by co-immunoprecipitation assays in Hs578T cells (Figure 1B).

To gain insights into the biological functions associated with the nuclear LOXL2 interactome, Gene Ontology (GO) analyses were performed using the DAVID functional annotation tool [37,38]. Functional enrichment analysis of molecular functions using the GOTERM_MF_DIRECT category revealed several distinct groups of functionally related proteins involved in RNA metabolism (Figure 2A). The most significantly enriched term that originated from the analysis was related to the RNA-binding cluster. Comparison of the interactome with the RNA-binding protein (RBP) database [39] showed that 86% of the proteins (93 out 107) of the LOXL2 interactome were in fact RBPs (highlighted in red in Figure 1A). Indeed, the LOXL2 interactome functional enrichment analysis of biological processes using the GOTERM_BP_DIRECT category revealed that LOXL2-interacting RBPs cover all facets of RNA metabolism, including splicing, modification, intracellular trafficking, translation, and decay (Figure 2B) [40]. Additionally, most of the LOXL2-interacting RBPs identified in this study have been associated with different cancer traits [41,42,43,44,45] and are being used as potential targets for cancer therapeutics [46]. The finding of these RBPs as potential partners for LOXL2 allows us to hypothesize the modulation of RBPs’ function as a new molecular mechanism of LOXL2’s action in tumor progression.

### 2.2. LOXL2 Interacts with RBPs through the SRCR-1 Domain

SRCRs are evolutionarily conserved domains of about 110 amino acids that were first discovered in the type I macrophage scavenger receptor [5]. There are two types of SRCR domains: class A and B, which are characterized by the presence of six and eight cysteine residues, respectively. The spacing pattern between the cysteine residues is conserved within the domains, and they participate in the formation of intramolecular disulfide bridges [47,48]. The function of SRCRs is not fully understood, but they are believed to mediate the binding of the SRCR superfamily of proteins to their substrates, which can be lipids, polysaccharides, or other proteins [5]. LOXL2 contains four SRCR domains of class A, and cumulative evidence suggests that some of LOXL2’s functions are independent of the amino oxidase catalytic domain but SRCR-domain-dependent [7,8,29,30,31,36,49,50].

To dissect the LOXL2 SRCR domain(s) involved in LOXL2’s interaction with RBPs, HEK293T cells were transfected with four HA-tagged LOXL2 mutants carrying a deletion of each of the individual SRCR domains (Figure 3A). ELAVL1 (also known as HuR) was one of the RBPs identified as a member of LOXL2’s nuclear interactome (Figure 1A); moreover, it constitutes one of the most intensively studied RBPs, and its association with tumorigenesis has been broadly reported [51]. Therefore, ELAVL1/HuR was selected to evaluate whether any of the LOXL2 mutants failed to bind this RBP. We performed co-immunoprecipitation assays using an antibody against HuR. The results (Figure 3B) showed that the mutant lacking the SRCR-1 domain (Δ-1) did not bind to ELAVL1/HuR. To expand this result to other proteins of the LOXL2 interactome, we performed LOXL2 and mutant Δ-1 co-immunoprecipitation assays with anti-HA antibody and analyzed the immunoprecipitates with antibodies against HSPA5, HNRNPC, MATRIN3, and HNRNPM. The results (Figure 3C) showed that the LOXL2 mutant devoid of SRCR-1 failed to bind to any of the evaluated proteins. 

We previously identified LOXL2 as a repressor of *E-cadherin* gene expression, and promoter assays indicated that the activity of the *E-cadherin* promoter was downregulated by LOXL2 through interaction with accessory proteins [29]. To analyze whether the transcriptional repression mediated by LOXL2 depends on its SRCR-1 domain, *E-cadherin* promoter assays were performed in the presence of wild-type LOXL2 or the LOXL2 mutants devoid of each individual SRCR domain. The results (Figure 3D) showed that repression of the *E-cadherin* promoter is abolished in the presence of the mutant deleted of the SRCR-1 domain, suggesting that the interaction of LOXL2 with auxiliary proteins to repress *E-cadherin* gene expression is mediated by the SRCR-1 domain in vivo. 

### 2.3. Identification of Putative RBP Candidates to Be Regulated by LOXL2 

Due to the considerable number of RBPs identified in this study, we found it necessary to introduce selective criteria that could guide us to specific RBPs participating in transcriptionally related LOXL2 function. We reasoned that if LOXL2 was modulating the activity of any RBP, this should reflect on the gene expression profile of cells devoid of LOXL2. For the same reason, many of the mRNAs from LOXL2-regulated genes should be binding targets of the RBPs. As a first approach to identify those specific RBPs targeted by LOXL2, we analyzed the effects of LOXL2 silencing on gene expression profiles and, subsequently, examined whether the genes regulated by LOXL2 were potential targets of the RBPs. To this end, we performed an RNA sequence analysis of the gene expression profile of Hs578T-shLOXL2 compared to control cells after confirming the efficient LOXL2 silencing at the mRNA and protein levels (Appendix A). The RNA sequence analysis revealed a total of 4738 significantly expressed genes (*p*-value adjusted by FDR ≤ 0.05) (Appendix A). Among them, we identified 621 differentially expressed genes with log2(fold change) ≥ 1 or ≤−1, the majority being downregulated in shLOXL2 cells (488 genes out of 621) (Figure 4A). 

None of the RBPs identified in the LOXL2 interactome exhibited altered levels in the RNA-Seq analysis. The data of the gene expression profile was confirmed by analyses of selected genes by RT-qPCR in Hs578T-shLOXL2 and MDA-MB-231-shLOXL2 cells compared to control cell lines. We observed that the pattern of gene expression was highly consistent between both cell lines (Figure 4B) and was comparable to the data obtained from the RNA-Seq analysis (Appendix A). 

To determine whether the DEGs belonged to a particular enriched pathway, we performed a gene set enrichment analysis (GSEA) using the “Hallmark” gene sets collection [52,53]. We observed several enriched pathways (*p*-value < 0.005) (Figure 4C), with EMT being one of the more significantly enriched signatures.

In fact, a search of the literature (Appendix A) revealed that there is experimental evidence of involvement in EMT for 19.3% of the genes (120 out 621) (Figure 5A). Ninety-two of the EMT-related genes were downregulated, while the rest were upregulated (Figure 5B).

Subsequently, we searched the ENCORI database [54] for the genes targeted by the RBPs identified in LOXL2’s interactome. We extracted the gene target list for the 24 RBPs presenting high-stringency CLIP-Seq data (CLIP-Seq data from ≥5 different experiments) [54]. Next, we compared the list of genes targeted by each RBP with the gene set of the EMT-related signature and the remaining DEGs (not EMT-related). In three cases (DDX3X, METTL3, and WTAP), we could not find any target among the DEGs. The remaining RBPs targeted 107 and 402 genes of the EMT-related (Appendix A) and non-EMT-related signatures (Appendix A), respectively, with different frequencies. The numbers of RBP hits in both signatures are shown in Appendix A. Fisher’s exact test analysis of the RBPs targeting genes in the EMT-related versus non-EMT-related signatures showed that the binding sites of six RBPs (ELAVL1, FMR1, IGF2BP1, TAF15, SRSF1, and U2AF2) were significantly enriched in the EMT signature (*p*-value < 0.05) (Figure 5C) (Appendix A), pointing to those RBPs as candidates to be regulated by LOXL2.

## 3. Discussion

The aim of this study was to identify new targets of LOXL2 that could be involved in the regulation of EMT. We performed LOXL2 co-immunoprecipitation experiments that revealed LOXL2’s interactions with a large number of RBPs. Additionally, we also found that the LOXL2–RBP interactions were mediated by the SRCR-1 domain. The low-resolution structure of LOXL2 has recently been determined [55], revealing that the SRCR-1–3 domains project linearly away from the catalytic domain, with the SRCR-1 domain being the most external one—a situation that is compatible with our finding that SRCR-1 domain could participate in protein–protein interactions.

RBPs play essential roles in modulating gene expression by regulating splicing, RNA stability, and protein translation, and we reasoned that the impact of LOXL2 on the function of specific RBPs may leave an imprint on the gene expression profile of cells lacking LOXL2. A similar approach has been successfully used to uncover the contribution of hnRNPC to chemotherapy resistance in lung cancer [56]. The analysis of the gene expression profile of breast cancer cells silenced for LOXL2 revealed 120 DEGs involved in EMT. Subsequently, we identified the RBPs significantly targeting the EMT signature. Despite the limitations of this study, the results obtained point to six RBPs (ELAVL1, IGF2BP1, FMR1, TAF15, SRSF1, and U2AF2) as candidates to be regulated by LOXL2 in the context of EMT. The evidence in support of this proposal is based on the implication of these RBPs in EMT and the possible modulation of their function by LOXL2-mediated lysine oxidation.

ELAVL1 (ELAV-like RNA-binding protein 1), also known as HuR (human antigen R), is a member of the Hu family of RNA-binding proteins. This family comprises four members: the neuronal proteins HuB, HuC, and HuD, and the ubiquitously expressed HuR [57]. ELAVL1 binds to AU-rich elements located in the mRNA 3′-UTR region and contributes to stabilizing and controlling the translation of mRNAs containing those motifs [58]. Thousands of mRNAs are ELAVL1 targets, including factors fostering diverse cancer traits such as proliferation (cyclins A2, B1, D1, and E1), angiogenesis (HIF1a, PTGS2, and VEGFA), survival (BCL2 and MCL1), and invasion (MMP9 and SNAI1) [59,60]. In addition, ELAVL1 has also been described as an EMT modulator in breast and pancreatic cancers [61,62]. Furthermore, several modifications of lysine residues impact on ELAVL1’s functions. For instance, ELAVL1’s protein stability is controlled by bTrCP1-mediated Lys 182 ubiquitination and MDM2-mediated neddylation of Lys 283, 313, and 326 [63,64]. In addition, neddylation of Lys 313 and 326 also affects ELAVL1’s RNA-binding activity [65]. Therefore, hypothetically, LOXL2 oxidization of those lysines could counteract ELAVL1’s ubiquitination/neddylation, thereby altering the half-life and/or RNA-binding capacity of ELAVL1, which could impact on its EMT-regulatory function.

IGF2BP1 (insulin-like growth factor-2 mRNA-binding protein 1) belongs to a conserved family of RBPs that comprises three members: IGF2BP1, -2, and -3. IGFBPs control the stability, export, and translation of different mRNAs, including KRAS and MYC mRNAs [66]. IGF2BP1 has been reported to induce EMT by enhancing the expression of *LEF1* and *SNAI2* [67]. Mechanistically, IGF2BP1 is activated by FBXO45 ubiquitination of Lys 190 and 450 [68], suggesting that the oxidation of those lysines by LOXL2 could block IGF2BP1’s ubiquitination and activation, thereby altering EMT.

FMR1 (fragile X messenger ribonucleoprotein 1) binds RNA, is associated with polysomes, and acts as regulator of translation [69]. When inactivated by a triplet nucleotide repeat expansion, it causes the neurodevelopmental disorder fragile X syndrome [70]. Despite its well-established role in the brain, in cancer cells FMR1 binds mRNAs involved in EMT and regulates their stability and translation [71]. FMR1’s stability is controlled by the ubiquitin ligase Cdh1-APC through lysine ubiquitination [72]. Prevention of lysine ubiquitination by LOXL2-mediated oxidation would prevent FMR1’s degradation and alter EMT status.

TAF15 (TATA-box-binding protein-associated factor 15) is a member of the TET family of proteins (which includes TLS, EWS, and TAF15). TAF15 (also known as TAFII68) was originally identified as a member of a subpopulation of the general transcription factor IID (TFIID) [73]. TFIID is the sequence-specific DNA-binding component of the RNA polymerase II transcriptional machinery. TFIID is composed of the TATA-binding protein (TBP) and at least 13 TBP-associated factors (TAF1-13). TAF15 is distinct from the other 13 best-known TAFs constituting the TFIID complex [74]. In this sense, TAF15 silencing does not affect general transcription but seems to impact a small subset of cell-cycle genes though miRNAs [74]. The TAF15/TBP complex is required for IL-6-activation-induced EMT and invasion [75]. Remarkably, LOXL2 is able to oxidize methylated lysines of TAF10—other components of the TFIID complexes—to represses TFIID-transcription-dependent genes [76]; therefore, it is conceivable that LOXL2 could exert a similar function on TAF15 to modify EMT status.

SRSF1 (RNA-binding protein serine/arginine splicing factor 1) is a splicing factor that, in addition to its function in alternative splicing, participates in nonsense-mediated mRNA decay, mRNA export, and translation [77]. Its involvement in EMT is well established as a regulator of alternative splicing of hundreds of genes, but recently it has been shown that it can induce EMT by stabilizing *RECQL4* mRNA [78]. SRSF1’s stability is also restrained by lysine ubiquitination [79], which could eventually be altered by LOXL2-mediated lysine oxidation.

U2AF2 (U2 small nuclear RNA auxiliary factor 2) is a component of the spliceosome complex that appears mutated with low frequency in myelodysplastic syndromes [80], but to the best of our knowledge its role in EMT regulation has not been tested. U2AF2’s stability is controlled by lysine ubiquitination [81]. In addition, JMJD6-mediated hydroxylation of Lys 15, 38, and 276 is required for the regulation of a large set of alternative splicing events [82]. Thus, lysine oxidation by LOXL2 could modify U2AF2’s function.

Altogether, our results suggest that LOXL2’s interaction with the candidate RBPs could alter cancer traits at different levels. LOXL2’s interaction with ELAVL1 or IGF2BP1 may modulate the mRNA stability of proto-oncogenes, cytokines, and growth factors to promote cell survival, metastasis, and drug resistance [83,84]. By interaction with FMR1, LOXL2 may modify the mRNA translation of genes implicated in the hallmarks of cancer, such as *TP53*, *VEGF*, *hTERT*, *TGFB2*, and other essential oncogenes [85]. LOXL2 can collaborate with TAF15 in the regulation of genes such as *CDKN1A*, *CDK6*, *CCND1*, and other cell-cycle genes at the posttranscriptional level to control cell proliferation [74]. LOXL2 may also alter alternative splicing through interaction with SRSF1, which, in turn, may affect apoptosis and proliferation [86]. Finally, LOXL2–U2AF2 interaction may also alter pre-mRNA splicing. U2AF2 is an essential component required for the splicing of vertebrate pre-mRNAs, and it has been described to mediate the alternative splicing of CD44, which confers metastatic potential to tumors [81].

Our LOXL2 interactome analysis sheds light on new functions in EMT regulation for this multifaceted protein. Nevertheless, the large number of RBPs identified in this study suggests that LOXL2 could be implicated in many facets of mRNA metabolism that could impact on multiple cell functions beyond the EMT process, and that deserve future investigations. Concerning the LOXL2–RBPs–EMT link, further research to characterize the functional interplay between LOXL2 and selected RBPs, along with its impact on EMT, will help to uncover novel roles of intracellular LOXL2 in tumorigenesis.

## 4. Materials and Methods

### 4.1. Cell Culture and Plasmid Constructs

Human HEK293T, MDA-MB-231, and Hs578T cell lines were obtained from the American Type Culture Collection. Cells were grown in DMEM medium (Gibco, Grand Island, NY, USA) supplemented with 10% fetal bovine serum, 10 mmol/L glutamine (Life Technologies, Carlsbad, CA, USA), and 1% penicillin/streptomycin (Invitrogen, Waltham, MA, USA). All cell lines were grown at 37 °C in a humidified 5% CO_2_ atmosphere. Cells were routinely tested for *Mycoplasma* contamination.

Human breast carcinoma Hs578T and MDA-MB-231 cells silenced for LOXL2 (shLOXL2) and control cells (shCTRL) were generated by stable transfection with pSuper-shLOXL2 and pSuper-shEGFP vectors, respectively, as described in [18]. Briefly, cells were transfected using Lipofectamine (Invitrogen) reagent with 5 μg of plasmids and selected in the presence of puromycin (1 μg/mL) for 2–3 weeks. LOXL2 silencing was confirmed by RT-qPCR and WB (Appendix A).

The human pcDNA3-Flag-LOXL2 vectors were as previously described in [87]. The human pReceiver-M06-HA-LOXL2 was purchased from GeneCopoeia (Ex-Y2020-M06). LOXL2 mutants carrying individual SRCR domain deletions (∆1–∆4) were generated from pReceiver-M06-HA-LOXL2 by site-directed mutagenesis and were performed by Mutagenex Inc. LOXL2 mutant ∆1 lacks amino acids 58 to 159, mutant ∆2 lacks amino acids 188–302, mutant ∆3 lacks amino acids 326–425, and mutant ∆4 lacks amino acids 435–544. The mouse *E-cadherin* promoter (−178 to +92) fused to the *luciferase* reporter gene was as previously described in [88].

### 4.2. Identification of Nuclear Proteins Associated with LOXL2

Nuclear protein extracts were obtained as described in [89]. Briefly, pcDNA3-Flag control and pcDNA3-Flag-LOXL2-transfected HEK293T cells were suspended in 10 mM HEPES pH 7.9, 10 mM KCl, 0.1 mM EDTA, 0.1 mM EGTA, 1 mM DTT, and a protease inhibitor cocktail (Roche). Cells were homogenized by adding 0.5% NP-40, and the tube was vigorously vortexed for 10 s. The homogenate was centrifuged for 5 min at 3000× *g*, and the supernatant containing cytoplasmic proteins was discarded. The nuclear proteins were extracted in 20 mM HEPES pH 7.9, 400 mM NaCl, 1 mM EDTA, 1 mM EGTA, 1 mM DTT, and a protease inhibitor cocktail on ice for 30 min with occasional gentle shaking. The nuclear extracts were centrifuged for 10 min at 12,000× *g*, and the supernatants were incubated overnight at 4 °C with 0.5 mg of Dynabeads Protein G (Thermo Fisher, Waltham, MA, USA) previously coated with 3 μg of anti-Flag antibody. The beads were washed three times with 20 mM HEPES pH 7.9, 1 mM EDTA, 1 mM EGTA, 1 mM DTT, and a protease inhibitor cocktail and then boiled in Laemmli sample buffer for 5 min. The supernatants were sent to the proteomic core facility of the Universidad Complutense de Madrid for protein identification by liquid chromatography–tandem mass spectrometry (LC-MS/MS).

### 4.3. Co-Immunoprecipitation

HEK293T cells were transiently transfected with pReceiver-LOXL2-HA and derivative mutants using Lipofectamine reagent (Invitrogen) with 2 μg of indicated plasmids. After 48 h, the cells were homogenized in 50 mM Tris-HCl pH 7.4, 150 mM NaCl, 5 mM EDTA, 1% Triton X-100 (IP buffer), and a protease inhibitor cocktail for 30 min at 4 °C with occasional gentle shaking. The homogenates were centrifuged for 15 min at 10,000× *g* and the pellet was discarded. Total lysates (0.5 mg) were incubated overnight at 4 °C with 3 μg of anti-ELAVL1 (Abcam, Cambridge, UK; ab136542) or anti-HA (Roche Life Science, Penzberg, Germany; 11-867-423-001) antibodies. Immune complexes were isolated by incubation with 1 mg of Dynabeads Protein G in IP buffer and a protease inhibitor cocktail for 3 h at 4 °C. After extensive washing (five times) with 1 mL of IP buffer, the immune complexes were eluted by incubation for 5 min at 95 °C in Laemmli sample buffer. Co-immunoprecipitation of endogenous LOXL2 in Hs578T cells and selected RBPs was performed as described above, except that the homogenate was incubated with anti-LOXL2 (Abcam; ab96233) or IgG antibody (Millipore, Burlington, MA, USA; cat. #PP64B). The proteins present in the eluted fractions were separated by SDS–polyacrylamide gel electrophoresis and analyzed by immunoblotting using an LOXL2 antibody from OriGene (TA807444); ELAVL1 (ab 200342), HSPA5 (ab21685), HNRNPC (ab133607), or MATRIN3 (ab151714) antibodies from Abcam; and HNRNPD (sc-166577) or HNRNPM (sc-20002) antibodies from Santa Cruz.

### 4.4. Promoter Assays

Luciferase reporter assays were performed as described in [90]. Briefly, transfections were carried out using Lipofectamine in the presence of 50 ng of empty pReceiver vector; pReceiver-LOXL2-HA, Δ1, Δ2, Δ3, or Δ4 expression vectors; 200 ng of mouse *E-cadherin* promoter; and 10 ng of pCMV-β-gal as a control for transfection efficiency. Luciferase and β-galactosidase activities were measured using the luciferase and β-Glo assay substrates (Promega) and normalized to the promoter activity detected in cells transfected with the empty pReceiver vector.

### 4.5. Gene Expression Profile Analysis

RNA from three independent clones of Hs578T-shCTRL control and Hs578T-shLOXL2 cells was used to perform RNA sequence analyses (Sistemas Genómicos, Valencia, Spain). Significant differentially expressed genes (DEGs) were selected by cutoff of *p*-values adjusted by false discovery rate (FDR) ≤ 0.05 and fold change greater than 2.

### 4.6. RNA Extraction, cDNA Synthesis, and Quantitative PCR (qPCR)

RNA was extracted and quality-tested in the Genomics Core Facility at the Instituto de Investigaciones Biomédicas Alberto Sols CSIC-UAM (Madrid, Spain). For cDNA synthesis, 1 μg of RNA was reverse-transcribed into cDNA using 200 units of M-MLV reverse transcriptase (Promega Corporation, Madison, WI, USA), 5 μL of M-MVL buffer (Promega), 0.5 μg of random primers (Promega), 10 mM dNTP mix (Bioron, Römerberg, Germany), and 25 units of RNaseOUT™ Recombinant Ribonuclease Inhibitor (Invitrogen) in a final reaction volume of 25 μL. Real-time qPCR was performed using Power SYBR Green Master Mix (Thermo Fisher) on the Applied Biosystems StepOne™ machine (Thermo Fisher). Each reaction was performed with 20 ng of cDNA and 9 pmol of specific forward and reverse primers (Appendix A). Values were normalized to the *GAPDH* housekeeping gene, and relative expression levels were analyzed by the 2^−ΔΔCt^ method. qPCRs were carried out in three independent samples assayed in triplicate.

### 4.7. Statistical Analysis

In the RT-qPCR assays, the *p*-values were generated using Student’s *t*-test (unpaired, 2-tailed); *p*-values < 0.05 were considered statistically significant. Data are presented as the mean ± standard error of the mean (SEM).

The analyses of RBP targets’ enrichment in the EMT signature with respect to the rest of the DEGs were performed by comparing the numbers of targets between both groups of genes using Fisher’s exact test. A *p*-value < 0.05 was considered statistically significant.

## 5. Conclusions

This study intended to identify nuclear partners of LOXL2 involved in EMT regulation. We report that LOXL2’s nuclear interactome is composed of numerous RBPs functioning in different aspects of mRNA metabolism. LOXL2 SRCR-1 was identified as the domain mediating interaction with RBPs. Analysis of mRNA changes associated with LOXL2 silencing, combined with the in silico identification of RBPs’ targets, pointed to six RBPs as candidates to be modulated by LOXL2, with likely influence on EMT regulation. Our results link LOXL2 to RBPs’ modulation in breast cancer; thus, using LOXL2 inhibitors in combination with RBP inhibitors may provide a more effective therapeutic strategy for treating breast cancer, as well as circumventing potential issues associated with targeting LOXL2 alone.

## Figures and Tables

**Figure 1 molecules-28-04433-f001:**
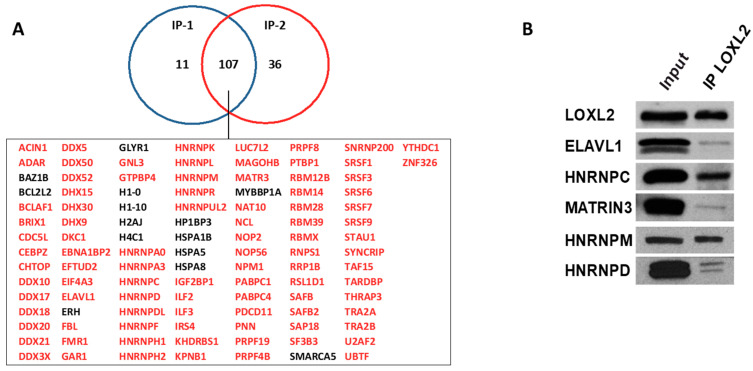
LOXL2’s nuclear interaction partners: (**A**) Immunoprecipitation of flag-LOXL2 in two independent experiments (IP-1 and IP-2) in HEK293T cells identified 107 proteins, which are listed below the Venn diagram. RBPs are marked in red. (**B**) Co-immunoprecipitation of endogenous LOXL2 in Hs578T cells and selected RBPs.

**Figure 2 molecules-28-04433-f002:**
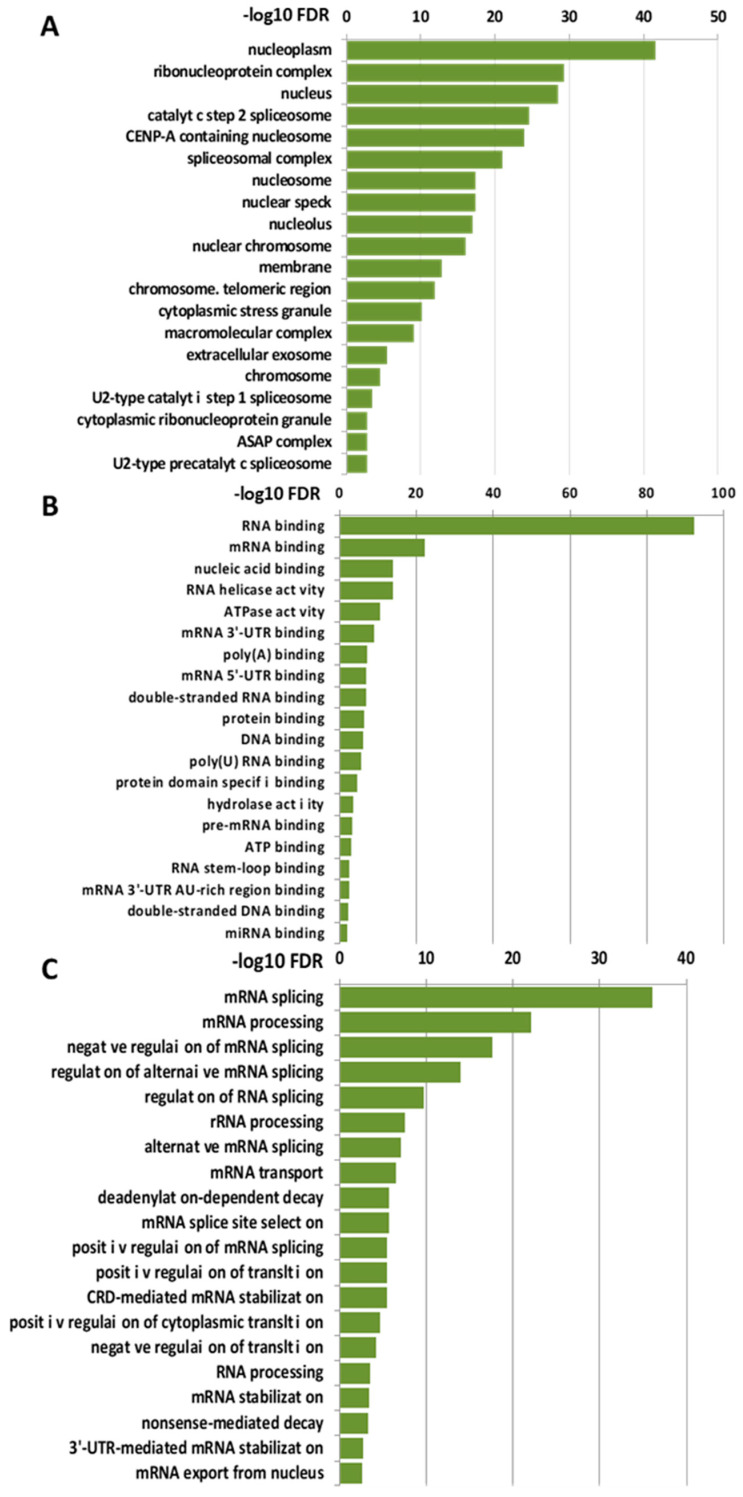
Molecular and biological functions of LOXL2’s nuclear interaction partners: Enriched categories were obtained using the DAVID functional annotation tool and ranked by the FDR value (−Log10 FDR) shown on the *x*-axis. (**A**) Functional enrichment analysis of molecular function using the GOTERM_CC_DIRECT category. (**B**) Functional enrichment analysis of biological function using the GOTERM_MF_DIRECT category. (**C**) Functional enrichment analysis of biological function using the GOTERM_BP_DIRECT category.

**Figure 3 molecules-28-04433-f003:**
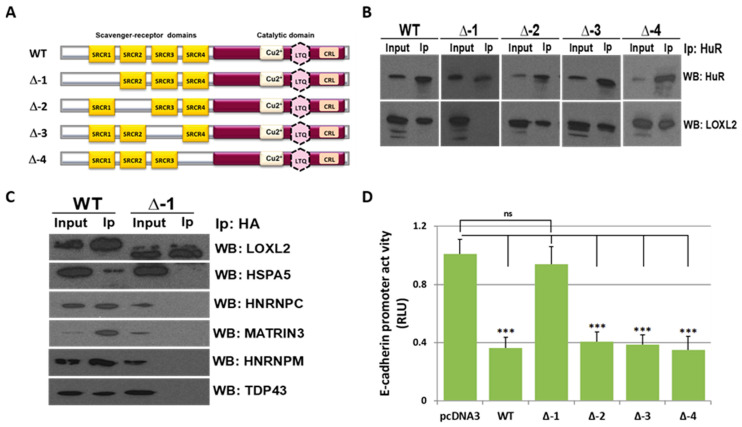
LOXL2’s SRCR-1 domain is required for protein interaction: (**A**) Schematic representation of wild-type LOXL2 (WT) and mutants carrying a deletion of the following SRCR domains: SRCR-1 (Δ-1), SRCR-2 (Δ-2), SRCR-3 (Δ-3), or SRCR-4 (Δ-4). (**B**) Whole-cell lysates of HEK293T cells transfected with LOXL2 WT or indicated SRCR-deletion mutants were immunoprecipitated with anti-ELAVL1/HuR antibody and analyzed by WB, with anti-LOXL2 antibody and anti-HuR antibodies as controls. (**C**) HEK293T cells were transfected with LOXL2 (WT) or the mutant carrying the deletion of the SRCR-1 domain (Δ-1); LOXL2 was immunoprecipitated using anti-HA antibody and analyzed by WB with the indicated antibodies. (**D**) The activity of the *E-cadherin* promoter in HEK293T cells was measured in the presence of the indicated LOXL2 mutants. The activity was determined as relative luciferase units (RLU) and normalized to the activity detected in the presence of the control pcDNA3 vector. Results represent the mean ± SEM. of at least three independent experiments performed in triplicate (*** *p* < 0.001, ns, not significant).

**Figure 4 molecules-28-04433-f004:**
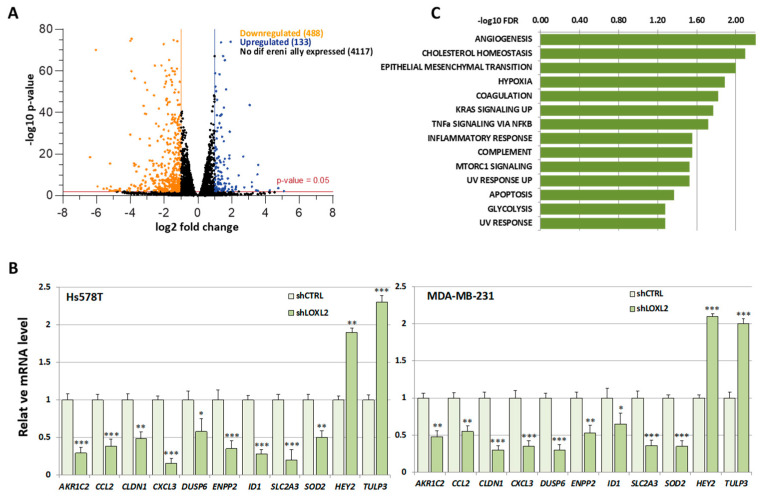
Gene expression profile of LOXL2-silenced cells: (**A**) Volcano plot showing transcriptome changes in LOXL2-depleted cells. The cutoffs were established at the log2 fold change > 1.0 or <−1.0 and the *p*-value < 0.05. Orange and blue dots represent significantly downregulated and upregulated genes, respectively. (**B**) Quantitative RT-qPCR confirming regulation of selected genes in LOXL2-ablated cells (dark green) compared to LOXL2 control cells (light green). Data are the mean ± SEM of three independent experiments assayed in triplicated samples in two independent breast cancer cell lines. The *p*-value was calculated by two-sided unpaired Student’s *t*-test. (**C**) GSEA plot of differentially expressed genes (DEGs) showing enrichment of hallmark signatures. FDR values (−log10 FDR) of such enrichments are shown on the *x*-axis. (* *p* < 0.05, ** *p* < 0.01, *** *p* < 0.001).

**Figure 5 molecules-28-04433-f005:**
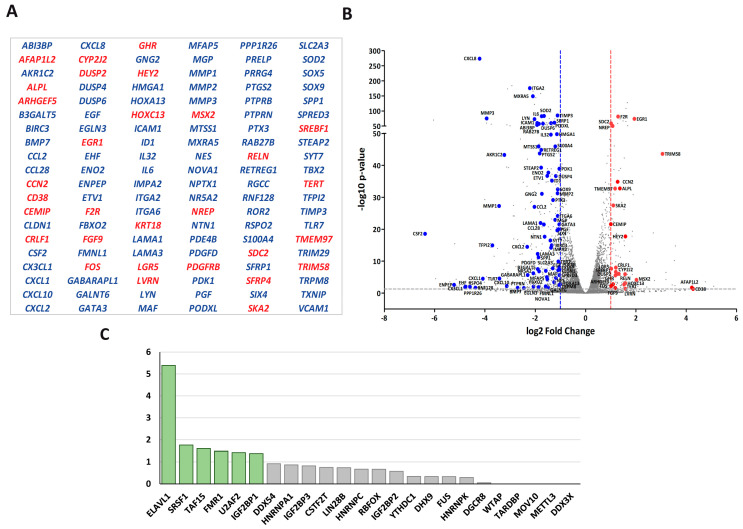
RBP candidates to be regulated by LOXL2: (**A**) List of differentially expressed genes in LOXL2-silenced cells with a described role in EMT. Up- and downregulated genes are marked in red and blue, respectively. (**B**) Volcano plot showing LOXL2-regulated EMT-related genes. The cutoffs were established at the log2 fold change > 1.0 (red line) or <−1.0 (blue line) and the *p*-value < 0.05 (grey line). Red and blue dots denote up- and downregulated EMT-related genes, respectively. (**C**) Enrichment plot of RBPs’ targets in the EMT-related signature relative to the non-EMT-related signature. The *p*-values were obtained by using Fisher’s exact test to compare the number of targets in both groups of genes. A *p*-value < 0.05 was considered statistically significant. Boxes marked in green represent significantly enriched RBPs.

## Data Availability

Not applicable.

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
