# Peer review of "New Functions of Intracellular LOXL2: Modulation of RNA-Binding Proteins"

_molecules, 2023, doi:10.3390/molecules28114433_

Round 1

Reviewer 1 Report

In this article, the authors focused on the interaction of LOXL2 with RBPs involved in several aspects of RNA metabolism. LC-MS/MS identified 107 proteins associated with LOXL2; most identified proteins were considered RBPs. Gene expression profile analysis combined with in silico identification of RBPs targets showed that six RBPs were candidates for substrates of LOXL2 action in the tumorigenic process. My specific comments are as below. I hope my words are beneficial for improving the article.

1. Lack of negative controls in Fig 1B, 3B, and 3C. The authors must show that the RBP is not detected in normal IgG or empty vectors in immunoprecipitation experiments to demonstrate that the binding between RBP and LOX2 is specific.

2. Although the authors have experimentally demonstrated that LOX2 binds to RBP, it is necessary to illustrate how this binding affects the function of RBP. For example, the process of HuR protein is regulated by intracellular localization and modification of itself. Of course, the detailed mechanism should be investigated in the future. However, the authors should explore the expression level, subcellular localization, and/or their modification of the identified RBP using knockdown and overexpressed cells.

3. Most of RBPs function by binding to their target mRNAs. Using the RIP method, the authors need to show whether changes in LOX2 expression alter the binding of the identified RBPs to their target mRNAs.

Reviewer 2 Report

- Figure 2 GO term enrichment analysis should include Cellular Component (CC) to confirm that Extracellular Region is indeed enriched, as LOXL2 is an extracellular protein.

- Figure 3 legend parenthesis "(***p,0.001" should be "(***p<0.001".

Reviewer 3 Report

In the manuscript “New functions of intracellular LOXL2: modulation of RNA binding proteins”, Eraso et al described new activity of Lysyl oxidase-like 2 (LOXL2), an extracellular enzyme involved in extracellular matrix remodeling. Through interactome analysis, the authors discovered the interactions of LOXL2 with different RNA binding proteins. Most of these RNA binding proteins are involved in several aspects of RNA metabolism, and might play critical role(s) with LOXL2 in the tumorigenic process. Coimmunoprecipitation of endogenous LOXL2 in Hs578T cells and selected RBPs clearly indicated their associations. In addition, they also identified that SRCR-1 domain in LOXL2 is required for interacting with these RNA binding proteins. This report is very easy to read, well written, interesting, and useful for further medical applications. Accordingly, this reviewer recommends publication. Minor revision is needed to sketch a signaling pathway for putative modulation of RNA binding proteins by LOXL2.

Reviewer 4 Report

The authors in the reviewed article entitled: New functions of intracellular LOXL2: modulation of RNA binding proteins investigated the role of lysyl oxidase-like 2 protein on matrix remodeling. First of all, the authors correctly noted that the above protein is involved in the first step of metastasis. This observation justified their study as attractive from the anticancer drug design point of view. Therefore, in the manuscript, the selected RBPs have been taken into consideration as LOX2 regulators. At this point, the authors made acceptable screening and statistical analysis of genes. Moreover, the experimental data have been also done: cell culture and plasmid constructs, identification of nuclear proteins associated to LOXL2, coimmunoprecipitation, promoter assays, gene expression profile analysis, RNA extraction, cDNA synthesis, and quantitative PCR (qPCR). This manuscript is interesting and well-written.

Before acceptance for publication, I expect that authors put some paragraph about new anticancer therapies and their target in the introduction part, extend the conclusion in the answer to the following question: why their discovery is important for the medicine? It would be a wonder if the materials and method part will be much more detailed.

Round 2

Reviewer 1 Report

The authors have satisfactorily addressed the issues raised in my previous comments, and I recommend publishing the paper in Molecules.